# The SAIL dataset of marine atmospheric electric field observations over the Atlantic Ocean

Susana Barbosa[1], Nuno Dias[1,2], Carlos Almeida[1,2], Guilherme Amaral[1,2], António Ferreira[1,2], António Camilo[3], and Eduardo Silva[1,2]

[1]INESC TEC - INESC Technology & Science, Porto, Portugal
[2]ISEP, Instituto Superior de Engenharia do Porto, Porto
[3]CINAV. Marinha, Lisboa, Portugal

**Correspondence:** Susana Barbosa (susana.a.barbosa@inesctec.pt)

**Abstract.**

A unique dataset of marine atmospheric electric field observations over the Atlantic Ocean is described. The data are relevant not only for atmospheric electricity studies, but more generally for studies of the Earth's atmosphere and climate variability, as well as space-earth interactions studies. In addition to the atmospheric electric field data, the dataset includes simultaneous measurements of other atmospheric variables, including gamma radiation, visibility, and solar radiation. These ancillary observations not only support interpretation and understanding of the atmospheric electric field data, but are also of interest in themselves. The entire framework from data collection to final derived datasets has been duly documented to ensure traceability and reproducibility of the whole data curation chain. All the data, from raw measurements to final datasets, are preserved in data repositories with a corresponding assigned DOI. Final datasets are available from the Figshare repository (https://figshare.com/projects/SAIL_Data/178500) and computational notebooks containing the code used at every step of the data curation chain are available from the Zenodo repository (https://zenodo.org/communities/sail).

## 1 Introduction

The atmospheric electric field is an ever-present feature of the Earth's atmosphere. Since the surface of the Earth and the ionosphere are good conductors, while the atmosphere is a reasonably good electrical insulator, an electric current flowing through the the Earth's atmosphere connects the Earth's surface to the ionosphere, constituting Earth's global electrical circuit (e.g Markson (2007); Rycroft et al. (2008); Williams (2009). The small density current flowing between the ionosphere and the Earth's surface is only of the order of a picoampere per square meter ($10^{-12}$ Am$^{-2}$) but it is able to produce a vertical electric field between 100 and 300 Vm$^{-1}$ near ground level (e.g. Burns et al. (2012)).

The global atmospheric electric field exhibits diurnal variability driven by the variation of global thunderstorm activity, influenced by the tropical distribution of land masses, above which thunderstorms preferentially form late in the day (local time). Non-lightning-producing storms (referred as electrified shower clouds) are also an important contribution to the global electric circuit, as proposed initially by (Wilson, 1921). Both thunderstorms and electrified shower clouds contribute to global

electric circuit through the descent of negative charge (Mach et al., 2010; Liu et al., 2010; Mach et al., 2011; Williams and Mareev, 2014).

The global nature of the Earth's electric field, and its diurnal variability, were confirmed by data collected in a series of campaigns aboard the *Carnegie* vessel between 1915 and 1929, showing that the electric field exhibits a diurnal variation, reaching its highest values at 19:00 UTC, regardless of the location on the globe (Parkinson, 1931; Torreson, 1946). This diurnal variation came to be known as the "Carnegie curve", and it is used to this day as the reference for the diurnal variation of the global atmospheric electric field (Markson, 2007; Harrison, 2013, 2020).

This diurnal feature of the global atmospheric electric field is hard to observe in non-marine measurements of the electric field, as it is usually hidden by local sources of variability including aerosols and particulates, ambient radioactivity, and anthropogenic influences such as power lines, electrical infrastructure and communication systems. Marine measurements of the atmospheric electric field are therefore very relevant for several atmospheric studies, but rare. Buoy measurements of the atmospheric electric field are becoming available (Wilson and Cummins, 2021), with the advantage of detailed monitoring for
prolonged periods of time at a specific location, though lacking the spatial distribution enabled by ship-based observations.

In a climate change context, the need for such observations of the atmospheric electric field over the ocean is even more compelling, as the electrical conductivity of the ocean air is clearly linked to global atmospheric pollution and aerosol content (Price, 1993; Rycroft et al., 2000; Harrison, 2004). Measurements from the research vessel *Oceanographer* in 1967 indicated values of atmospheric electrical conductivity consistent with the original Carnegie observations in the remote South Pacific, but
a decrease over the Atlantic of at least 20%,which was attributed to an increase in North hemisphere aerosol pollution (Cobb and Wells, 1970). Here we present a unique dataset of atmospheric electric field measurements performed over the Atlantic Ocean in the scope of the SAIL (Space-Atmosphere-ocean Interactions in the marine boundary Layer) project (Barbosa et al., 2023c). Section 2 gives an overview of the monitoring campaign and methodology for collecting the data, section 3 describes the dataset and applied quality assurance procedures, and concluding remarks are provided in section 5.

## 2   Monitoring campaign

The SAIL monitoring campaign started on January 5th 2020 aboard the Portuguese navy ship *NRP Sagres* (Figure 1) for an initially planned circum-navigation expedition of 371 days. However, the voyage was interrupted due to the covid pandemic, and the campaign was thus restricted to the Atlantic Ocean. Figure 2 depicts the ship's trajectory during the SAIL campaign. After a short stop at Cape Town for provisions, the ship departed the same day back to Portugal, having arrived in Lisbon on
May 10th, after a required technical stop for repairs at Cabo Verde.

The monitoring system of the SAIL campaign is described in detail in Barbosa et al. (2022). In brief, the atmospheric electric field and ancillary variables are measured on the mizzen mast of the *NRP Sagres* ship (see Figure 1) and transmitted to a dedicated on-board computer. Every measurement is tagged with a timestamp with microsecond precision based on the system clock in coordinated universal time (UTC). The system clock is corrected by a PPS (Pulse Per Second) signal available
from a Global Navigation Satellite System (GNSS) receiver.

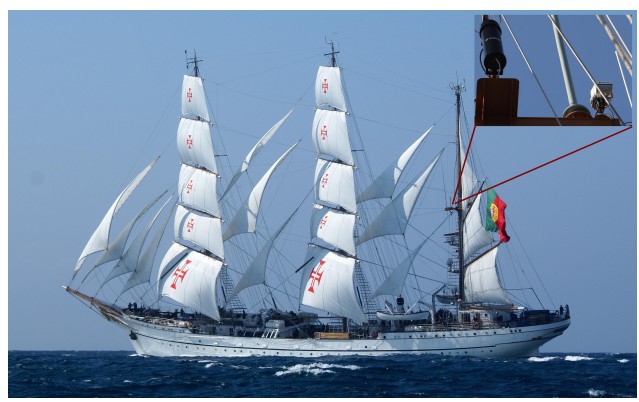

**Figure 1.** Photo of the *NRP Sagres* ship in full sail; the inset shows the location on the mast of the gamma radiation sensor (black cylinder, left) and of the primary electric field CS-110 sensor (right).

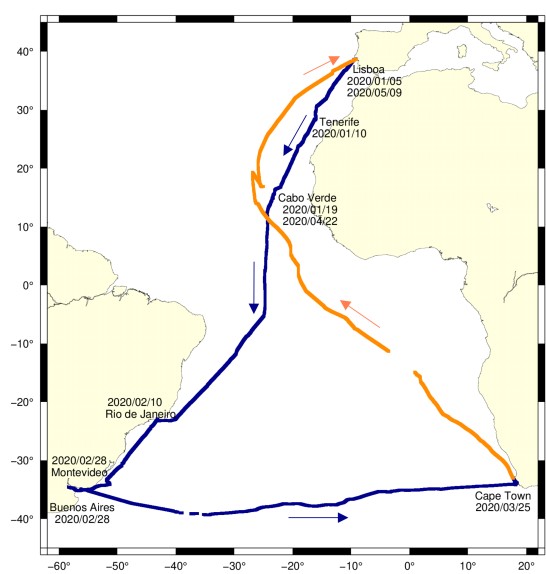

**Figure 2.** Trajectory of the *NRP Sagres* ship from January to May 2020; blanks correspond to periods with no data.

The atmospheric electric field is measured near the top of the ship's mast, at about 20 meters height, with an automatic electric field meter sensor CS-110 (Campbell Scientific, UK) measuring the vertical component of the electric field by means of of an oscillating grounded shutter. A secondary measurement is performed at the same mast but closer to the ship deck, at a height of around 5 meters, using an identical instrument. Ancillary atmospheric variables are measured close to the main electric field sensor, at the 20 meters height, and include gamma radiation, visibility, and short-wave solar radiation. Gamma radiation resulting from natural radioactivity, including the radioactive decay of radon gas progeny, and from the interaction of cosmic rays and atmospheric gas molecules, is a direct source of atmospheric ions. Ions influence cloud and aerosol processes (Harrison and Carslaw, 2003) and changes in ion concentration and/or ion mobility impact the local atmospheric electric field by changing atmospheric conductivity (Harrison and Tammet, 2008). Visibility and solar radiation are used to assess atmospheric conditions, as weather conditions causing changes in charge distribution or ion mobility influence the local atmospheric field (e.g. Bennett and Harrison (2007)). Atmospheric conductivity decreases with increased aerosol concentration (e.g. Kamsali et al. (2009)),which in turn decreases visibility, as higher aerosol loads scatter and absorb more light. Therefore the presence of aerosol implies both the reduction of the air's electrical conductivity and the visual range (Brazenor and Harrison, 2005; Harrison, 2012) .

Gamma radiation is measured with a $76 \times 76$ mm$^2$ NaI(Tl) cylindrical scintillator (Scionix, the Netherlands) equipped with an electronic total count single channel analyzer measuring total counts of gamma radiation in the 475 keV to 3 MeV energy range, which is optimal for the detection of radon progeny (Zafrir et al., 2011). Possible sources of the measured gamma radiation include gamma rays from the radioactive decay of potassium-40 in seawater and from gamma-emitting radionuclides in the uranium and thorium series, typically present in suspended sediments at the ocean surface and attached to atmospheric aerosols. Cosmic radiation contributions are expected to be comparably much smaller since the secondary cosmic radiation reaching the earth's surface is composed by only about 2% of gamma radiation (Wissmann et al., 2007). The scintillator is encased in a water-proof container protecting it from the harsh marine conditions and installed next to the electric field instrument (starboard side), in an upright position and pointing upwards, in order to have the field of view of the instrument towards the atmosphere above, rather than encompassing the ocean surface and the ship itself. Visibility is measured at the port side with a visibility sensor SWS050 (Biral, UK) providing meteorological optical range measurements in the range from 10 m to 40 km. Short-wave solar radiation is measured next to the electric field sensor using incoming (Apogee, SP-510) and outgoing (Apogee, SP-610) solar radiation sensors. Local meteorological information (rain, atmospheric pressure, temperature, and wind) is manually recorded by the ship's crew every 1 hour as part of the navy's operations routine (meteorological information is not recorded when the ship is in port).

During the 126-days of the SAIL campaign, all measurements were performed continuously at a rate of 1Hz, except for visibility with measurements every 1-minute. Overall data completion is $> 95\%$. Data loss due to malfunction of the monitoring system occurred on 8th and 9th March (during the trip from Buenos Aires to Cape Town) and then from 4 to 6 April (in the leg from Cape Town to Lisboa), due to issues on the onboard computer and storage systems. The missing segments in the ship's trajectory represented in Figure 2 correspond to these data gaps. The data management strategy for all the data collected in the scope of the SAIL campaign is detailed in the SAIL data management plan (Barbosa and Karimova, 2021).

## 3 Data and quality assurance

All the data from the SAIL campaign are preserved in order to foster their reuse in different scientific domains and to enable initially unforeseen uses of the data. All data handling processes are fully documented to ensure traceability and reproducibility.

The raw campaign data (Barbosa et al., 2021) are only available upon request due to its large size (around 700 GB). This dataset of raw measurements includes the data obtained directly from the ship on-board system (designated as ship data), the data (designated as sensor data) obtained from the ship data by correcting logging errors (Amaral and Dias, 2021) and the data (designated as geosensor data) obtained from the sensor data by adding two additional columns corresponding to latitude and longitude based on the GNSS data from the campaign (Ferreira, 2021). The logging errors are caused by non-deterministic communication failures between the instrument and the main onboard computer, as well as occasional power shortages, which result in parsing errors due to incomplete lines and non-standard characters in the output files. Such errors are corrected automatically by in-house developed software that checks the correct number of fields in each line and and removes the line if it does not match the expected count. (Amaral and Dias, 2021) .

Except for the meteorological observations, all data were collected continuously during the SAIL campaign, thus including both measurements performed over the ocean as well as coastal measurements taken when the ship was docked in port during its various stops along the journey (see Figure 2). To facilitate the usage of the data for studies requiring ocean-only observations (e.g. Barbosa et al. (2023b)), a flag denoting fully-ocean days, when the ship is away from the coast, is added to the final datasets (Figure 3).

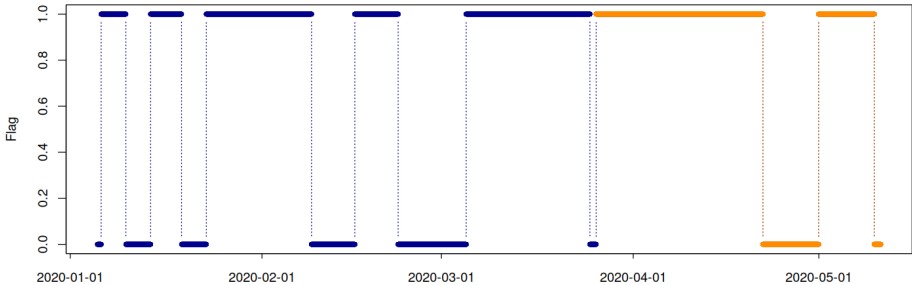

**Figure 3.** Flag distinguishing fully ocean (=1) and fully or partially land (=0) days for the measurements taken between January $5^{th}$ and May $9^{th}$ 2020. The same colours as in Figure 2 are used for the first leg of the ship trajectory (blue) and for the returning leg (orange).

Pre-processed data (Barbosa et al., 2023a) are produced from the raw data by implementing quality-control and pre-processing procedures. These procedures and the resulting quality-assured derived datasets are described in section 3.1 for the atmospheric electric field data, and in section 3.2 for the ancillary data.

### 3.1 Atmospheric electric field

Measurements of the atmospheric electric field are performed with no site-specific corrections. The default value of 0.1 provided by the CS-110 manufacturer for the site calibration factor, $C_{site}$, of a sensor with the shutter at 2 m above flat ground

(Campbell Scientific, Inc., 2023), is used both for the primary instrument and the secondary (lower) one, designated as E1 and E2, respectively. The behaviour of the two instruments is addressed in section 3.1.1.

The raw atmospheric electric field data are first pre-processed for basic quality-control (section 3.1.2). Corrections are applied at a subsequent stage, and are fully documented, in order to be able to trace back all the steps to reproduce and/or to further modify the data processing (section 3.1.3). Selection of fair weather atmospheric electric field data is described in section 3.1.4.

### 3.1.1 Zero-field measurements

The two electric field instruments were factory-calibrated before the SAIL campaign, and further evaluated after the campaign in terms of zero-field measurements, by using a zero field cover plate attached to the instrument's shutter in its typical downward-facing orientation, enabling to ground any electric field that would be measured by the instrument, and thus assess its potential contamination. The data were collected on land, at the same height of about 2m, over three consecutive days (June 3 to 5, 2022). Figure 4 summarises the zero-field electric field measurements and Figure 5 the corresponding leakage current measurements. These results indicate that the primary electric field sensor has a smaller error and lower leakage current than the secondary sensor, but both sensors perform well, the difference to zero being below 4 V/m and leakage currents below 0.025 nA.

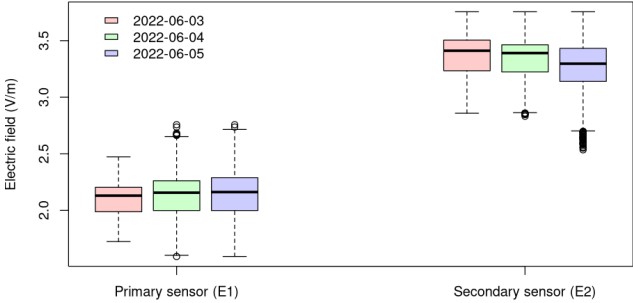

**Figure 4.** Boxplots of zero-field electric field measurements. The lower limit of each box corresponds to the 1st quartile of the values, the upper limit to the 3rd quartile, and the horizontal line inside each box represents the median of the data. The vertical whiskers extend to 1.5 times the interquartile range (3rd quartile minus 1st quartile), and values outside that interval are represented as circles.

### 3.1.2 Atmospheric electric field data pre-processing

Pre-processing of the raw atmospheric electric field data is documented in Barbosa (2023c), and includes:

– checking the instrument status code; if different than 1 (indicating good instrument health) the corresponding measurement is set as missing (flagged as NA);

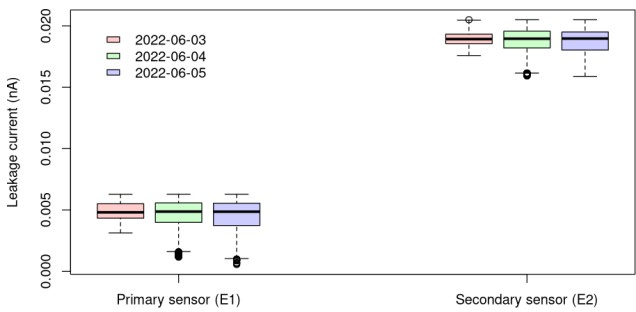

**Figure 5.** Boxplots of zero-field leakage current measurements. Same conventions as for Figure 4.

- changing the sign of the atmospheric electric field measurements to comply with the sign convention denoting the potential gradient as positive under undisturbed atmospheric electrical conditions (e.g. Harrison and Nicoll (2018)) since the electric field is downward-directed in fair weather conditions;

- averaging 1-second electric field measurements into 1-minute values;

- averaging geographical coordinates (taking into account angularity) to 1-minute averaged values;

- computing the standard deviation every 1–minute from the 1-second measurements;

- checking the record continuity and inserting a flag (NA) for missing times in order to ensure a continuous time series of atmospheric electric field observations.

The pre-processed dataset obtained by applying these procedures to the raw data (but before application of the corrections that will be described in section 3.1.3) is available from the INESC TEC data repository (Barbosa et al., 2023a).

Figure 6 presents examples of 1-minute pre-processed electric field observations from the two sensors for days with contrasting weather conditions. These examples emphasise the consistency of the temporal variability of the electric field measurements from the two sensors, on one hand, and on the other hand the large difference in the corresponding values of the atmospheric electric field, with values from the secondary instrument (Figure 6b) substantially lower and less variable than the ones of the primary instrument (Figure 6a) . These differences are not explainable by differences in the performance of the two instruments (see section 3.1.1) nor by differences in the height of the sensors, as these would not explain the reduced variability of the secondary electric field measurements. Plausibly the differences between primary and secondary electric field observations result from the location of the secondary sensor and consequent field distortion effects. While the primary sensor, near the top of the mast, has relatively unimpeded surroundings, the secondary (lower) sensor is adjacent to several structures of the ship, likely distorting the local electric field. Despite this difficulty the secondary electric field measurements, at the lower height, are kept in the dataset, but their use and interpretation should be cautious, particularly in terms of absolute values.

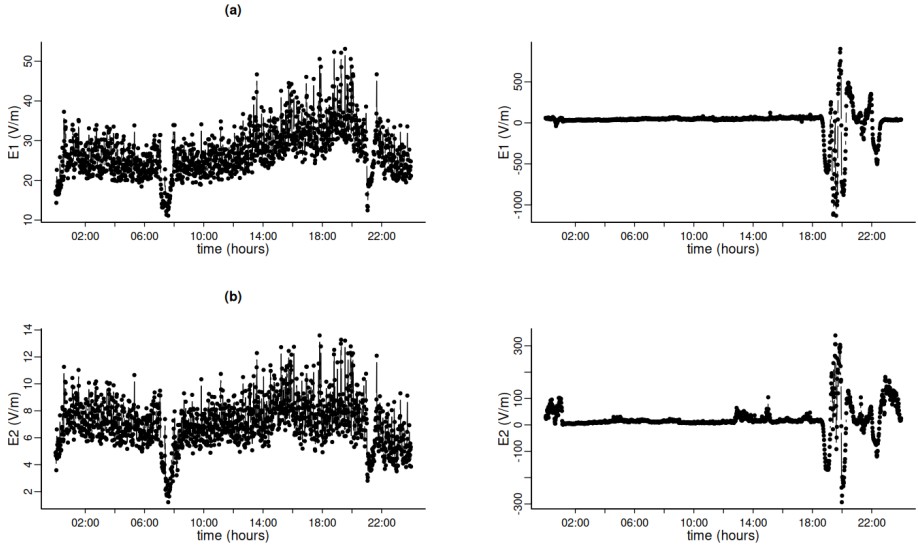

**Figure 6.** Examples of pre-processed electric field observations for a clear day (on February 2nd, left) and for a rainy day (on January 28th, right), from the primary (higher) instrument (top) and the secondary (lower) instrument (bottom).

### 3.1.3 Atmospheric electric field data corrections

The atmospheric electric field measurements taken on the ship depend on the location of the electric field sensors and are influenced by the ship's geometry. Although the site-related field distortions do not influence relative variations of the atmospheric electric field, they impact absolute values. Quantification of site specific influences is a challenging task. A first attempt to address the difference between measurements due to the location of the sensor on the top mast relative to on-shore measurements is presented in section 3.1.3(a). The differences between the primary electric field sensor, located at the top of the mast, and

the secondary sensor, located further down the mast, are addressed in section 3.1.3(b) .

### (a) Correction of primary electric field measurements

The influence of the height at which the primary atmospheric electric field measurements are performed is assessed by considering simultaneous observations of the atmospheric electric field conducted at the height of about 20 meters near the top of the mast (using instrument E1) and at sea level (standard 2 meters height from the ground), with the secondary instrument (E2)

placed on shore when the ship was docked at the Lisbon Naval Base (Figure 7). Due to logistic and operational constrains, the measurements were performed for a short period of about 2 hours on June 16th 2020, under fair weather conditions. These simultaneous measurements are presented in Figure 8. The pier measurements exhibit several spikes, which are absent in the mast measurements, likely resulting from human activity at the pier disturbing the electric field measurements. The temporal variability of the two measurements is consistent, with a Pearson's linear correlation coefficient of $0.848$, but there is a clear

bias between the mast and the pier measurements, the mast measurements being significantly lower (averaging 68 V/m ) than

the pier measurements (which average 119 V/m). The bias is estimated by means of a linear model, represented in Figure 9. The (positive) correlation between the two measurements is statistically significant ($[0.84, 0.85]$ is the $95\%$ confidence interval) and the fitted linear model has a slope equal to 1.76 ($\pm 0.013$), explaining $72\%$ of the variance. The linear model's intercept is zero (statistically not significant). These estimates are used for height correction of the primary measurements of the atmospheric electric field on the mast, by multiplying all the mast observations by 1.76: $E1_{h\_corr} = E1 \times 1.76$ (V/m).

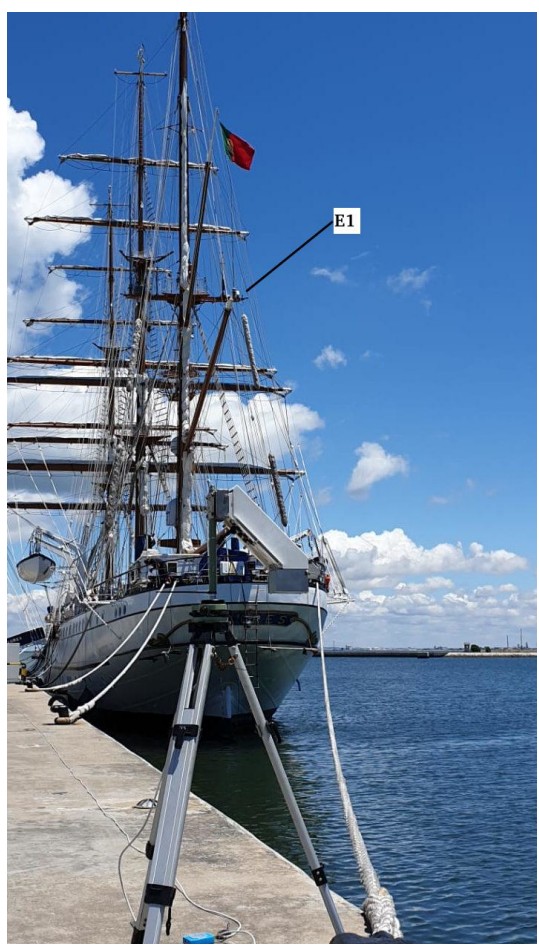

**Figure 7.** Photo showing the instruments used for the simultaneous measurements of the atmospheric electric field at the mast and on shore.

### (b) Correction of secondary electric field measurements

Figure 10 summarises the height-corrected primary electric field observations and the secondary electric field measurements in terms of its daily median values (Figure 10, right) and in terms of daily median differences $E1_{h\_corr} - E2$ (Figure 10, left). The differences are in general positive (primary measurements larger than secondary electric field measurements), averaging 56 V/m. This bias estimate is used to correct secondary electric field observations: $E2_{corr} = E2 + 56$ (V/m).

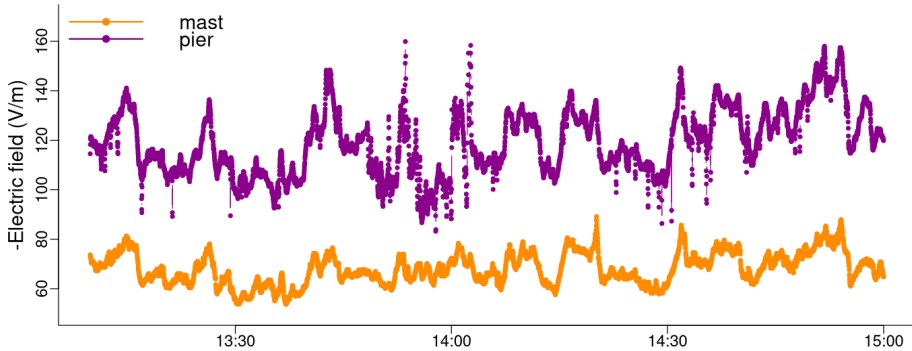

**Figure 8.** Time series of simultaneous atmospheric electric field measurements every 1-second performed at the mast of the ship (at a height of about 20 meters) and at the pier in Lisbon Naval Base (at the standard height of 2 meters) under fair weather conditions in 2020-06-16.

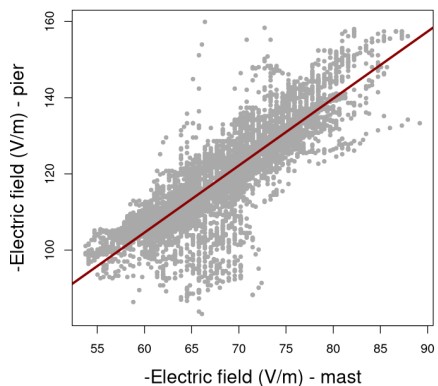

**Figure 9.** Scatterplot and fitted linear model for the observation represented in Figure 8.

The datasets of height-corrected primary electric field observations and bias-corrected secondary electric field observations are available from the Figshare repository (Barbosa et al., 2024b). The datasets include the time stamp (in the format yyyy-mm-dd HH:MM:SS), the1-minute averaged potential gradient in V/m after applying the corrections described above, the corresponding standard deviation in V/m, longitude, latitude, and the flag signalling whether it's a fully ocean day (=1) or a fully or partially land day (=0).

### 3.1.4 Atmospheric electric field data selection

A dataset of selected atmospheric electric field observations is derived from the dataset of primary corrected electric field observations by applying the following data-driven criteria:

– Non-negative Potential Gradient values (corresponding to 98.6% of the observations);

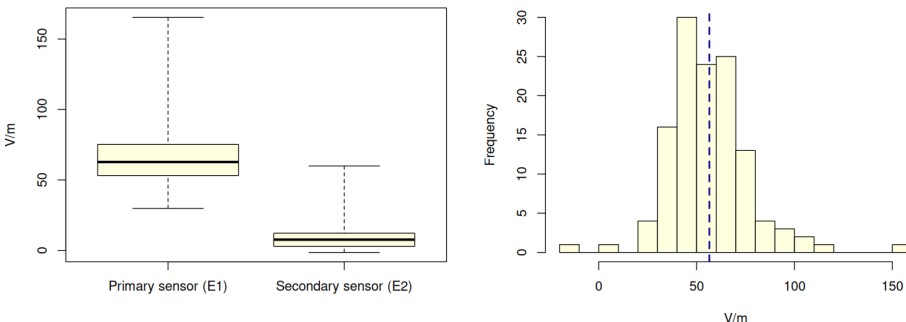

**Figure 10.** Daily median values of height-corrected primary electric field observations and secondary electric field measurements (left) and corresponding daily median differences $E1_{h\_corr} - E2$ (right), the dashed vertical line representing the average of the differences.

- Observations flagged as a fully-ocean day (see Figure 3) which correspond to 71.9 % of the observations.

In addition to these criteria, the following fair weather criteria (Harrison and Nicoll, 2018) are applied based on the available ancillary and meteorological information (see section 3.2):

- Dry day, according to manual precipitation records (corresponding to 85.8% of the days);

- Clear sky (meteorological optical range $\geq 30,000$ meters), a condition fulfilled by 60.1 % of the observations.

The application of these criteria results in retaining 35.6 % of the corrected primary electric field observations. The resulting dataset of these fair weather marine observations of the atmospheric electric field is available from the Figshare repository (Barbosa et al., 2024c).

Figure 11 shows the hourly boxplots for the selected fair weather electric field observations displaying the median value (horizontal solid line) and the 1st and 3rd quartiles of the observations (lower and upper vertical limits of the box, respectively). The hourly values were computed by averaging the 1-minute electric field observations for each preceding 59 minutes. The Sagres data display the typical *Carnegie curve* shape, with minimum around 04:00 UTC and maximum around 19:00 UTC, but the amplitude of the curve represented by hourly median values, of only about 18 V/m, is substantially smaller, corresponding to $30\%$ of the amplitude of the *Carnegie curve*.

## 3.2 Ancillary observations

### 3.2.1 Gamma radiation

Pre-processed gamma radiation data are obtained from the raw data by aggregating (adding) the gamma radiation counts measured every second into 1-minute values, calculating average geographical coordinates every 1-minute, and by checking data continuity and flagging missing measurements, which correspond to $4.4\%$ of the time series values. Further quality-control is performed by inspecting the pre-processed 1-minute data, and identifying anomalous values, typically sharp spikes (lasting

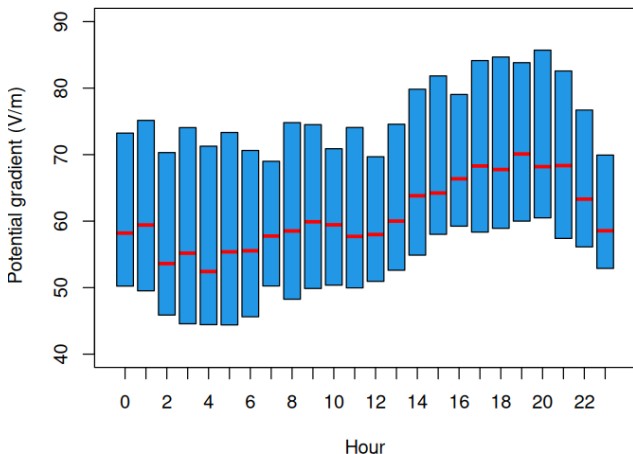

**Figure 11.** Hourly boxplots (1st to 3rd quartile) of SAIL fair weather atmospheric electric field observations. The horizontal red line represents the hourly median value of the potential gradient.

less than 3 minutes), and anomalously low values before/after a data gap (associated with recovery of the instrument after power failure). These outliers (1.2% of the time series values) are set as missing, as exemplified in Figure 12. The jupyter notebook (Granger and Pérez, 2021) implementing these pre-processing and quality-control steps is preserved in the Zenodo repository (Barbosa (2025c)). The resulting dataset of quality-assured gamma radiation observations is available from Figshare (Barbosa et al., 2025a).

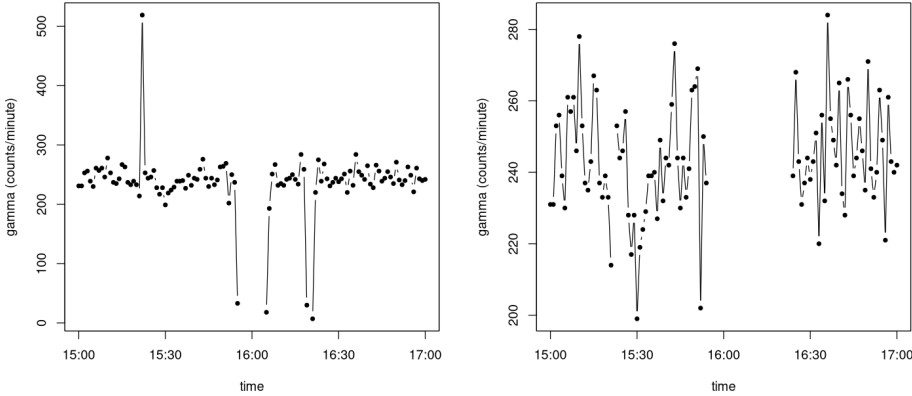

**Figure 12.** Example (16th January 2020) of pre-processing of 1-minute gamma radiation observations: spikes and anomalously low values before/after a data gap (left) are set as missing (right).

 ### 3.2.2 Visibility

Pre-processed data are obtained by extracting meteorological optical range measurements from the raw visibility data and then checking temporal continuity and inserting a flag (NA) for missing observations, in order to produce a continuous time series (Barbosa, 2024). The quality-assured time series of meteorological optical range observations is available from the Figshare repository (Barbosa et al., 2024a).

The meteorological optical range measured by the visibility sensor reflects the transparency of the atmosphere, and is an useful parameter to assess local atmospheric conditions. As an example, Figure 13 displays the visibility data for a clear day and for a rainy day. In the first case visibility values are high and at the upper limit of the instrument's range, except for cloudy conditions reducing visibility around 08:00, while in the latter case visibility values are low, with lowest observations around 17:00 and 19:00, associated with rain episodes.

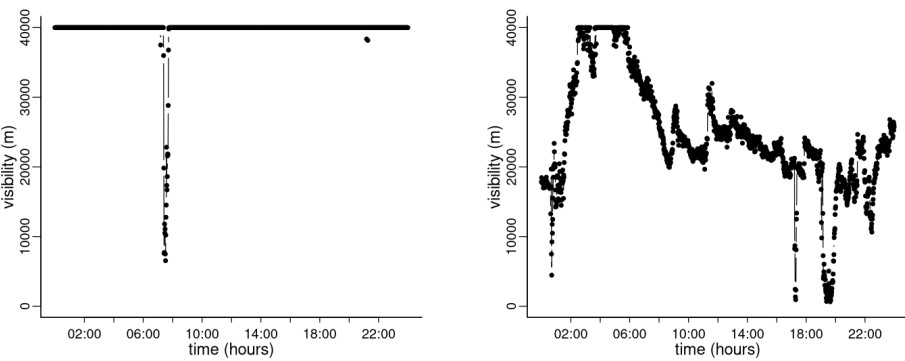

**Figure 13.** Example of visibility observations for a clear day (on February 2nd, left) and for a rainy day (on January 28th, right).

### 3.2.3 Solar radiation

Raw solar radiation data every 1-second are pre-processed to produce 1-minute averaged incoming and outgoing short-wave solar radiation. Inspection of the data for quality-control reveals the existence of non-valid negative values of solar radiation. These negative (and small magnitude) values of solar radiation are replaced by zero. Inspection of the incoming solar radiation data for each hour of the day reveals a few small values during night hours, which are set as zero. A much larger number of 230 non-zero night values is found in the case of outgoing radiation - likely reflecting the effect of the ship's own illumination - and these values are set as missing. The jupyter notebooks implementing these quality-control procedures are preserved in the Zenodo repository (Barbosa, 2025b). The resulting quality-assured datasets of incoming and outgoing short-wave solar radiation are available from the Figshare repository (Barbosa et al., 2025b).

    Figure 14 displays an example of the daily variability of 1-minute incoming solar radiation observations for the same days 235 as in Figure 13. For the sunny day the diurnal pattern is more regular and incoming solar radiation values are higher. It must be

noted that although the solar radiation sensors were installed high on the mast, some partial shading and/or enhanced reflection by the ship's sails cannot be discarded.

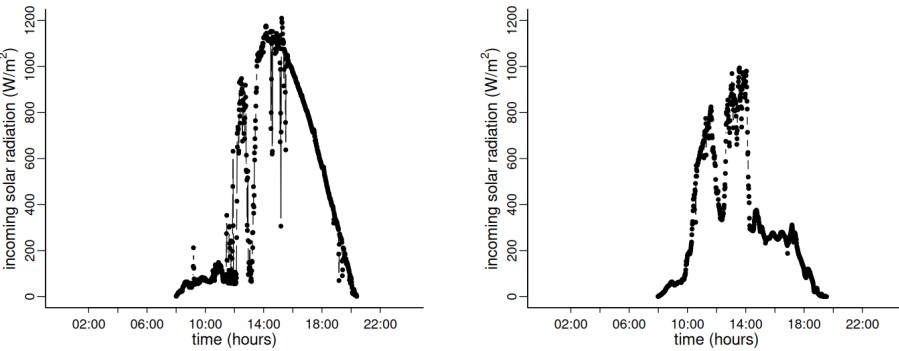

**Figure 14.** Example of incoming short-wave solar radiation observations for a clear day (on February 2nd, left) and for a rainy day (on January 28th, right).

### 3.2.4 Meteorological information

Local meteorological information is collected every hour by meteorological observers of the ship's crew (Table 1). The raw data (Camilo, 2021) were corrected by homogenising non-standard missing values flags and by removing headers and formatting features in order to enable further automatic processing. The resulting corrected data (Barbosa, 2023b) are subject to further quality-control procedures specific to each meteorological parameter, as detailed in the jupyter notebook made available in the Zenodo repository (Barbosa, 2023a). These include, in addition to removal of obvious outliers, the translation of visibility classes from Portuguese to English based on WMO-No. 471 (WMO, 2018), and the homogenisation and translation of qualitative precipitation information. The resulting quality-assured dataset of meteorological observations is available from the Figshare repository (Barbosa and Camilo, 2023).

**Table 1.** Meteorological data over the Atlantic Ocean collected onboard the NRP Sagres ship during the SAIL campaign.

| Datafile column | Meteorological variable | Unit / format |
|---|---|---|
| 1 | Date | yyyy-mm-dd |
| 2 | Time | HH:MM, local time |
| 3 | Latitude | DD° M.M |
| 4 | Latitude | suffix (N or S) |
| 5 | Longitude | DDD° M.M |
| 6 | Longitude | suffix (E or W) |
| 7 | QNH (Query Nautical Height) | mbar |
| 8 | Temperature - dry bulb | °C |
| 9 | Temperature - wet bulb | °C |
| 10 | Dew point | °C |
| 11 | Relative humidity | % |
| 12 | Water temperature - bucket | °C |
| 13 | Water temperature - hull | °C |
| 14 | True wind direction | ° |
| 15 | True wind speed | knots |
| 16 | True wind force | beaufort scale |
| 17 | Wave direction | compass half-wind |
| 18 | Wave height | m |
| 19 | Visibility | qualitative code [1] |
| 20 | Cloud cover | oktas |
| 21 | Precipitation | qualitative code [1] |

[1] excellent, very good, good, moderate, poor

[2] moderate, light, drizzle, drizzle moderate, drizzle light

## 4 Code and data availability

All the code and data is publicly available. The project SAIL community on Zenodo (https://zenodo.org/communities/sail/) contains the technical documents related to the SAIL data, and the computational (jupyter) notebooks used at the different stages of data processing (Table 2). Raw data (Barbosa et al. (2021), DOI: 10.25747/b2ff-kg31) and pre-processed data (Barbosa et al. (2023a), DOI: 10.25747/58P6-6B76) are available from INESC TEC RDM repository. Final datasets (Table 3) are available from the Figshare repository, under the SAIL data project (https://figshare.com/projects/SAIL_Data/178500).

250

**Table 2.** Code (Jupyter notebook) available on the project SAIL community on Zenodo (https://zenodo.org/communities/sail/).

| Title | DOI | Reference |
|---|---|---|
| Pre-processing and quality-control of of electric field data | 10.5281/zenodo.10276613 | Barbosa, 2023c |
| Pre-processing and quality-control of gamma radiation data | 10.5281/zenodo.14803667 | Barbosa, 2025c |
| Pre-processing of visibility data | 10.5281/zenodo.11621789 | Barbosa, 2024 |
| Pre-processing and quality-control of solar radiation data | 10.5281/zenodo.14720715 | Barbosa, 2025b |
| Pre-processing of meteorological data | 10.5281/zenodo.10150266 | Barbosa, 2023a |
| Computational notebook for the figures in this paper | 10.5281/zenodo.14833426 | Barbosa, 2025a |

**Table 3.** Datasets available on the SAIL data project on Figshare (https://figshare.com/projects/SAIL_Data/178500).

| Title | DOI | Reference |
|---|---|---|
| Atmospheric electric field data | 10.6084/m9.figshare.19692391.v1 | Barbosa et al., 2024b |
| Fair weather atmospheric electric field data | 10.6084/m9.figshare.26022001.v1 | Barbosa et al., 2024c |
| Gamma radiation data | 10.6084/m9.figshare.20393931.v4 | Barbosa et al., 2025a |
| Visibility data | 10.6084/m9.figshare.19692394.v3 | Barbosa et al., 2024a |
| Solar radiation data | 10.6084/m9.figshare.24614754.v2 | Barbosa et al., 2025b |
| Meteorological data | 10.6084/m9.figshare.24613869.v1 | Barbosa and Camilo, 2023 |

## 5 Conclusions

The SAIL dataset of marine atmospheric electric field observations over the Atlantic Ocean is a unique dataset, relevant not
only for atmospheric electricity studies, but more generally for studies of the Earth's atmosphere and climate variability, as
well as space-earth interactions studies.

In addition to the atmospheric electric field measurements, the data presented here include simultaneous measurements of
other atmospheric variables, including gamma radiation, visibility, and solar radiation. These ancillary data not only support
interpretation and understanding of the atmospheric electric field observations, but are of interest in themselves (e.g. Barbosa
et al. (2023b)), as data seldom measured over the ocean, and even more rarely at the spatial and temporal resolutions achieved
in the SAIL campaign.

The measurement of the atmospheric electric field on a tall ship has several challenging aspects, including the variable site
geometry, particularly related to the changing configuration of the sails, and field distorting effects due to the ship's structures.
Corrections have been provided according to the best available information, but further simultaneous measurements on the ship
mast and on shore, away from the ship's (and other structures) influence, are clearly desirable. Another possibility to increase
confidence on the correction of the atmospheric electric field measurements would be the development of an electrostatic model
of the ship's geometry enabling to simulate deviations in the electric field due to local geometric and conductive influences.

Finding the correct reduction factor to adjust the ship observations for the variations introduced by the ship itself was already challenging during the Carnegie cruises (Hewlett, 1914; Torreson, 1946), and continues to be so in modern-day measurements. The absolute values provided for the atmospheric electric field need therefore to be taken with caution. Enhanced confidence is ensured by relative atmospheric electric field values.

The entire framework from data collection to final derived datasets has been duly documented in order to foster reproducibility of the whole data curation chain, and enable alternative data processing strategies and different corrections to be seamlessly implemented.

A follow-up monitoring of the atmospheric electric field aboard the NRP Sagres ship is currently ongoing, and corresponding datasets will be updated in a future effort.

*Author contributions.* SB: conceptualization, data curation, formal analysis, writing - original draft; ND, GA, AF: set-up of monitoring system, data collection; data curation; CA: set-up of monitoring system, data collection; AC, ES: resources, supervision.

*Competing interests.* The authors declare absence of competing interests.

*Acknowledgements.* The support provided by the NRP Sagres's crew and the Portuguese Navy is gratefully acknowledged. Project SAIL received funding from the Portuguese Ministry of Environment and Energy Transition through Fundo Ambiental protocol no 9/2020 and by the Portuguese funding agency, FCT - Fundação para a Ciência e a Tecnologia, within project UIDB/50014/2020. DOI 10.54499/UIDB/50014/2020 https://doi.org/10.54499/uidb/50014/2020.

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
