# Peer review of "The SAIL dataset of marine atmospheric electric field observations over the Atlantic Ocean"

_Earth System Science Data, 2024_

## Author Response (AR1)

**Response to referee comments**

We are grateful to all reviewers for their evaluation of our work and for the helpful corrections, comments and suggestions. Our point-by-point response to the reviewer comments is provided below, followed by a summary of changes implemented in the revised version.

Point-by-point response to referee comments

**Reply to referee comments – RC1**

Thank you, the mistake on line 222 is corrected in the revised version of the manuscript
* * *
**Reply to referee comments – RC2**

Susana Barbosa has devoted valuable attention to making scientific observations from a Portuguese Navy ship. I this work the emphasis lies with the measurement of the fair-weather electric field. At a time of resurging interest in the monitoring of the global electric circuits, these efforts are most welcome. The work definitely deserves to be published, but some suggestions come to mind for improving the final submission. These substantive issues are followed by detailed comments/edits on the text.
Summary: Publish after major revision

Substantive Issues:

Sources for the DC global circuit
Thunderstorms are given emphasis (page 1, line 14) as the source for the global circuit, but recent work by Mach et al. (2009, 2010) have substantiated the prescient suggestions of CTR Wilson (1921), a paper cited by the authors, that electrified shower clouds may play an equally important role. It is recommended that the authors study the relevant sections of Wilson (1921) and then modify their Introduction slightly.

We agree with the referee comments and suggestions, and the introduction was modified accordingly in the revised version of the manuscript.

Absolute calibration of electric field
The one aspect of this work in most need of additional attention is the absolute calibration. And this is never a trivial task. The E field magnitudes in Figures 5, 6, 8 and 9 are low. The Carnegie Curve" field is low by roughly a factor of two. This reviewer needs clarification of procedure. The very best thing to have done is to place the second sensor "on shore" but perhaps more rigorously than what is reported in line 136 on page 6, to mount the CS-110 flush-mounted on a large planar surface on the dock and well out of influence of the mast and rigging of the ship. (If flush-mounted the makers of the CS-110 have a sound procedure for absolute calibration.) Then one makes simultaneous recordings on shore (i.e., the flat dock) and on the ship mast in fair weather conditions to get the best form-factor for the ship's installation. The shore measurement is the absolute reference.

The procedure described above is exactly what members of the Carnegie Institution did with the calibration of the E field on the Maude, though instead of using a flat dock, they used the flat Arctic ice sheet in the vicinity of the ship, when it was locked in the ice.

Now if this rigorous procedure has not been undertaken, I am not quite sure what to suggest. One could adjust the mean value in Figure 6 to match the Carnegie ship measurements, or more recent ones undertaken by Wilson and Cummins (2021) on measurements made (also with CS-110 instruments) on buoys at sea off the coast of Florida.

Wilson, J. G., & Cummins, K. L. (2021). Thunderstorm and fair-weather quasi-static electric fields over land and ocean. Atmospheric Research, 257, 105618. https://doi.org/10.1016/j.atmosres.2021.105618

We agree with the referee's comments, absolute calibration is indeed the main issue that we have with the atmospheric electric field observations, and one that is not easy to address properly given the limitations we face in terms of logistics and operational procedures. Unlike the Carnegie ship, Sagres is not a dedicated research vessel but a ship of the Portuguese Navy, currently serving as a training, diplomatic and military ship. The SAIL monitoring campaign is being carried out within the ship's regular missions and military duties, which significantly constrains the timing and type of activities that we can undertake. The attempt that was made of making simultaneous measurements "on shore" and on the ship mast was certainly far from ideal, but even such an arrangement - ensuring the simultaneous availability of the ship on dock, authorisation for accessing the ship, support from the crew for work on the mast, and as much as possible fair weather conditions - was already quite challenging. The monitoring campaign is continuing, and we aim make every effort to achieve a solution to this problem, by replacing the simultaneous measurements on the mast and on dock by simultaneous measurements at the coast, on a plain beach site, and on the mast with the ship anchored near the coast, or eventually simultaneous measurements at the mast and on a smaller ship nearby.

We prefer not to adjust the electric field observations to the conventional values from the Carnegie measurements or modern available measurements. Despite careful and detailed calibration efforts the observations from Wilson & Cummins (2021) are on average 33% lower than the Carnegie observations. We opted instead to introduce in the revised version of the manuscript further cautionary notes on the reliability of the absolute values and on the limitations of the correction procedure, while pursuing further efforts to improve the calibration of the Sagres observations.

 Neither adjustment may be appealing to the authors, so I am hoping the more rigorous treatment has been pursued, or could still be pursued.  The mean values of 119 V/m reported in line 140 (page 7) is close to the Carnegie mean value of 130 V/m and in Figure 7 I see an eyeball mean near 120 V/m.  So now I wonder if this normalization was applied to the analysis in Figure 9, why isn't the mean value roughly twice what is shown here?
I do think something can be done to improve the situation overall.

The mean value reported in page 7, and that can be seen in Figure 7, refers to the mean value for the specific period represented in Figure 6 – overall (considering the full monitoring period) the average value is indeed substantially lower. The correction factor was applied to the analysis in Figure 9. The figure below shows the same plot as in Figure 9 before applying the correction (left) and after applying the correcting factor (right), confirming that the values would be even lower if not applying the stated correction factor.

[Figure]

[Figure]

Please note that while performing this check we noticed that the plot in Figure 9 was based on an earlier version of the data (in which different criteria were applied in the fair weather data selection procedure). Although the differences are not substantial, the figure was corrected in the revised version of the manuscript. Further detail on the boxplots and how the calculations were performed were added to the revised manuscript. The code used to produce all the figures in the manuscript was included in the revised version (link to the computational notebook added to Table 2).

Spikes in Figure 6
Isn't it likely that the spikes in Figure 6 in the purple record for the pier are due to cultural activity on the pier near the CS-110 that could not be completely suppressed?

Yes, it's likely that the spikes are due to disturbances induced by sporadic human activity at the pier. This was commented in the revised version of the manuscript.

Selection of common time series for two scenarios
I think data plots of the kind shown in Figure 6 transformed to a scatterplot in Figure 7 is a a good way to calibrate one field recording against another. I would suggest however the calculation of a formal correlation coefficient between the two records as further quantitative evidence that you are measuring (mostly) a common E field imposed on both instruments.

We agree, the formal correlation was computed and added in the revised version of the manuscript.

Confusion about data "while navigating" and "in port"
Here I cited line 70 on page 3 and line 86 on page 4. In Figure 2, does the "land" flag mean being in port and not under sail ("navigating". Please clarify all in the text.

We agree with the need for clarification, the term "navigating" was not intended to distinguish between being under sail or not, but being in port, the term was corrected in the revised version of the manuscript.

Energy bandwidth of gamma ray source
The information provided in line 63 on page 3 is relevant to an earlier interest on the reviewer's part in seeing whether electron runaway may have been occurring during precipitation events at sea. However, the rather low min value for gamma ray energy (475 keV) will allow for radon daughter products and that could dominate any event. To be more certain about real electron runaway, this lower threshold would be need to be substantially higher. See also Chilingarian (2018).

Chilingarian, A. (2018). Long lasting low energy thunderstorm ground enhancements and possible Rn-222 daughter isotopes contamination. Physics Review D, 98, 022007. https://doi.org/10.1103/physrevd.98.022007

The range of energies of the gamma sensor was indeed selected to focus on radon progeny observations (this information was added to the revised version of the manuscript), and we do not have any sensor with a higher minimum threshold… therefore the current measurements are not appropriate to detect such effects during precipitation events at sea, but the point is worthy considering if we manage to add a different type of sensor to the monitoring set-up in the near future.

Detailed comments/edits on the text:

Line 14   Add "and electrified shower clouds"
The introduction was updated, electrified shower clouds are explicitly mentioned in the revised version.

Line 22 "variation of global thunderstorm activity"
Done.

Page 23 delete "throughout the Earth"
Done.

Line 24   change "at the end of the day" to "late in the day"
Done.

Line 27 "came to be known"
Done.

Line 28 Can also cite Markson (BAMS, 2007)
Done.

Line 31 "The need for such observations…"
Done.

Line 34 Interesting. I had forgotten about this. That should be checked again sometime, maybe by this group.
Agree!

Line 41 The inclusion of a photo of the ship with mast and rigging would be helpful here.
Photo added in the revised version of the manuscript.

Line 44 "arrived in Lisbon"
Done.

Line 48 Why is the microsecond precision so important for your endeavors here?
It is really not important for the specific measurements we are considering, but it doesn't hurt...

Line 53 Suggest change from "rotating" to "oscillating"
Done.

Line 53 "at a height"; No problem of E field shielded by conductive rigging?
Changed. E field shielding by conductive rigging cannot be excluded...

Line 56 I think you need more details about "radon gas progeny" (the same suggestion I made for your other recent manuscript).
More details on the sources of gamma radiation were added to the revised version of the manuscript (section 2)

Lines 59-61 Think you need to tie together better the visibility variable and the conductivity variable, with aerosol being the key physical linkage.
We agree with the suggestion, and corresponding text was added in the revised version of the manuscript.

Line 65 Clarify why "pointing upwards"?
The motivation for the positioning of the sensor - to have the field of view of the instrument towards the atmosphere above, rather than encompassing the ocean surface and the ship itself – was added to the revised version of the manuscript (section 2)

Line 68 Is the outgoing also shortwave, or rather longwave radiation?
The sensor is designed to measure shortwave outgoing radiation.

line 74 change "voids" to "missing segments"
Done.

line 78 "to foster their reuse"
Done.

line 86 This seems contradictory with line 70 on the previous page.
Clarified in the revised version.

Figure 2 caption should tell what total time was involved.
The information was added to the caption in the revised version of the manuscript.

Line 94  2 meter height over flat terrain? (Your procedure is not entirely clear to me.)
The text was updated for clarification in the revised version of the manuscript.

Lines 102-103   Your procedure is not completely clear to me
Further details were added in the revised version

Line 106   4 V/m is a small fraction of typical fair weather fields
The low value shows no contamination of the instruments (as the measurements were taken with the zero-field cover)

Figure 3: minor DC offsets

Figure 4:  only Campbell (CS-110) people will know how to interpret these results (and me!)
Agree ;) we think nevertheless that this information should be available for the users of the dataset

Line 112   Suggest adding: "The electric field is downward-directed in fair weather conditions"
Done.

Line 118  "applying these procedures to the raw data"
Done.

Line 133  This suggests that only height is included but it is more complicated than that I think.
True, further information on the possible influences that would need to be corrected (other than simply height) was added in the revised version of the manuscript, at the beginning of section 3.1.3. The title of the subsection was changed to "Correction of primary electric field measurements".

Lines 135-136  Use of photos here would be helpful.
A photo was added in the revised version of the manuscript.

Line 138  Calculation of correlation coefficient would be helpful here.
Added in the revised version of the manuscript.

Line 141 Quantify the correlation.
Added in the revised version of the manuscript.

Figure 6 caption:  Add a statement about the weather condition for this short period of data.
Added in the revised version of the manuscript.

Figure 7   What is the origin of the points beneath the main linear scatter?  It would be useful to give the overall correlation coefficient.
These points, as shown by the red dots in the following figure, are mostly related to the period shortly before 14:00 for which the pier measurements show spikes and more erratic behaviour, likely due human activity at the pier, and the mast measurements are very stable, showing little temporal variability.

[Figure]

Line 150 What are the details of the height-corrected data?
Details added in the revised version of the manuscript.

Line 156 "as a fully-ocean day"
Done.

Line 162 Are "marine observations" all ocean days?
Yes, these are observations from fully-ocean days and fair weather selected according to the criteria detailed above. The text was slightly modified in the revised version in an attempt to make it clearer.

Line 166 Add a sentence about the quantitative comparison: the Carnegie curve amplitude variation is +/- 15% as I recall.
Added in the revised version of the manuscript.

Figure 9 is a very valuable result, but it would be nice if the absolute field were more accurate (see earlier remarks in item (2) above.
Agree! Unfortunately we do have an issue with absolute values, that we tried to make more explicit in the revised version of the manuscript.

Line 172 This reminds me of the spike in Figure 6 (top record). What do you attribute that to?
The spike in figure 6 appears only on the shore measurements, and not the mast ones, and is likely due to human activity on the pier, in the vicinity of the ground level instrument.

Line 174 What is the "jupyter notebook"?
Reference added in the revised version of the manuscript.

Line 175 "what is "Zenodo"?
Text updated in the revised version of the manuscript.

Figure 10 What do you think causes the spikes? Could they be chunks of radioactivity coming over? The first author is well-suited to answering this question.
Such isolated spikes in gamma radiation as shown in Figure 10 are rare (in total we have identified only about 20 of such isolated, single spikes, specifically on 2020-01-14, 2020-01-16, 2020-01-18, 2020-01-24, 2020-01-30, 2020-02-07, 2020-02-20, 2020-02-25, 2020-03-12, 2020-03-15, 2020-03-26,2020-04-02, 2020-04-07, 2020-04-09, 2020-04-10, 2020-04-15, 2020-04-16, 2020-05-06, and 2020-05-07). We do not have an explanation for them, but they are not found in the corresponding electric field measurements, as shown in the figure below for the same day and time period as in Figure 10. The period with no data due to power failure is visible as for the gamma data, but no apparent spike on the atmospheric electric field observations.

[Figure]

Line 184 "are consistently high"  They are constant.  Discuss that limit (40 km) (see line 67 on page 3)
Text updated in the revised version of the manuscript.

Figures 11 and 12  Why not show solar radiation for the same two days shown for visibility?  This would help point up the internal consistency of the overall data archive.
The solar radiation was shown for the same two days as for visibility, the apparent disturbances in solar radiation in the morning of the clear day are likely the result of partial shading by the ship's sails.

Line 204  "jupyter notebook"???
The reference for jupyter notebooks, the computational files describing the data analysis workflow, was included in the revised version of the manuscript.

Line 213 "include"
Done.

Line 222 "duly" ???
Corrected.

Section 5  Should this follow the Conclusions, or rather appear in an Appendix?
Thank you, the code and data availability section was moved before the Conclusions section.

What did the Portuguese Navy get out of the useful project?
The project served as a stepping stone and strengthened the collaboration between the Portuguese Navy and INESC TEC in the area of autonomous systems and environmental monitoring.
* * *
**Reply to referee comments – RC3**

This data paper presents and summarises the data obtained during the SAIL voyage. As the authors indicate, this dataset is important for long term monitoring of atmospheric electricity which, in turn, is important for long term environmental change. The data and how they were obtained are described fully and carefully, though I have added some minor suggestions below that I hope could improve clarity.

As the authors suggest, the electric field measurements appear reliable, but look likely to have been influenced by the ship's geometry. The field mills are calibrated in that they give readings in Volts per metre, and the relative variations in the data are still valid, for example the Carnegie-like curve in figure 9. However, without some sort of quantification of the reduction factor to account for the geometrical screening from the ship, comparisons of the absolute values are almost impossible. There seem to be two ways to work out the geometric factor to "reduce" the data to an idealised situation. The first approach would be to carry out more measurements with the sensors on the ship, which is probably no longer possible. Another approach, which may be more suited to retrospective analysis, is to use an electrostatic model of the ship geometry to work out the distortion at the position of the sensors.

This paper is worthy of publication without this reduction factor, which may take some time to evolve (as I believe it did for the original Carnegie voyages), but the authors do need to discuss this issue more than they do in the manuscript I saw.

We totally agree with the remarks on the needed reduction factor for the atmospheric electric field measurements. We have added additional information to the revised version of the manuscript regarding the limitations of the corrections provided for the measurements, improving Section 3.1.3. Additionally, we further discuss the limitations of absolute values of the atmospheric electric field in the conclusions section.

Minor comments

L 24 clarify that the timings are in local time
Done.

L30 might be good to spell out what you mean by these (eg aerosol) since you talk about pollution later
Added in the revised version of the manuscript

L82 please add a brief description of what the logging errors were and how they were corrected so the reader does not have to refer to the reference
Added to the revised version of the manuscript (beginning of section 3).

Figure 2 please spell out the colour scheme again so the figure can be understood without looking at Figure 1
The colour scheme was switched relative to Figure 1 and it was corrected – thanks!! The two tones of blue colour scheme was modified for clarity in the revised version of the manuscript.

L95 it isn't clear what you mean about the "default value of the sensor" when they were at different heights, please reword.
Rephrased and updated in the revised version of the manuscript.

Figs 3 and 4 please define box, whisker and outlier criteria
Done.

Fig 5 please label plots (a) (b) etc and refer to them in the text. It looks like 12 is missing from the y axis of the E2 plot bottom left.
Done. The figure was updated.

Fig 6 list date and location of these measurements in caption for completeness
Done.

Fig 8 Is the daily median difference plot a Gaussian distribution? I would expect it to be, but the plot suggests it might be a little skewed. Could the authors please do a statistical test for "normality" and discuss the origin of any "non normality"? Or repotting the histogram so that data looks more Gaussian might also help.
We replotted the histogram, making the histogram bins smaller. The daily median differences do seem a little skewed, but statistically it is a normal distribution – at least one cannot reject normality based on a statistical test, for example the well-known Shapiro-Wilk statistical test yields a p-value of 1.4e-05. Because normality tests have typically low power, I prefer to look at a quantile-quantile plot of the data, shown below (left plot).

The plot shows some deviations to what would be a "perfect" normal distribution (the diagonal line), particularly for higher values (corresponding to the spikes in the pier observations), but the deviation from the normal is acceptable. To give an idea of what would be the expected spread from the diagonal in a finite sample of normal values of the same size as the data we have, the left plot shows the same quantile-quantile plot but for a sample of simulated values from a normal distribution with the same mean and standard deviation as the median differences. Although the deviations are smaller in tat case – after all, these are values simulated from a perfect normal distribution, some spread in the tails still occurs, and tends to reduce by increasing the sample size.

[Figure]

L190 is it not more rigorous to replace any negative values with NA rather than assuming they are zero?
Yes, it is indeed less rigorous, therefore we have reprocessed the incoming solar radiation data, replacing the negative values by NA instead of zero. Figure 12 was updated with the new version of the data. The solar radiation datafiles on the data repository and the jupyter notebook containing the preprocessing code were also updated.

Table 1 should be "dry bulb" and "wet bulb" temperature
Corrected in the revised version of the manuscript.
* * *
**Reply to referee comments – RC4**

The preprint manuscript detailing the SAIL dataset of marine atmospheric electric field observations over the Atlantic Ocean is a significant contribution to the field of atmospheric science. It provides a comprehensive dataset that is well-documented and made publicly available, which is a strength of the study. The authors have taken care to ensure the traceability and reproducibility of the data curation chain, which is essential for such datasets to be useful in the scientific community. The manuscript is well-structured, and the findings are presented clearly, making the paper accessible to a broad audience. The interdisciplinary relevance of the dataset, spanning atmospheric electricity, climate variability, and space-Earth interactions, is well-articulated.

But I am concerned about the following points:

Major:

Data Interpretation and Context: While the dataset is extensive and valuable, the manuscript could benefit from a more detailed discussion on the implications of the observed diurnal variability of the atmospheric electric field. Specifically, how do these observations compare with existing models of the global atmospheric electric circuit, and what new insights do they provide regarding the influence of thunderstorm activity on this circuit?
Ancillary Data Analysis: The manuscript mentions ancillary measurements such as gamma radiation, visibility, and solar radiation. It would be beneficial to include a more detailed analysis of these variables in relation to the electric field data. For instance, are there any correlations between gamma radiation spikes

and changes in the atmospheric electric field, and if so, what might be the atmospheric or climatic implications?
Comparison with Other Studies: The paper references previous studies and observations, such as the Carnegie curve. It would enhance the manuscript if the authors could include a comparison of their findings with these historical data, discussing any deviations and potential reasons for them.
Minor:

We agree with the relevance and importance of all the three points mentioned above. The only reason they are not addressed in the current manuscript, is that they are out of scope of the current data paper. Specifically, ESSD aims & scope indicate that "Any interpretation of data is outside the scope of regular articles." We expect that the dataset will be used not only by our group but also by other researchers, originating further subsequent papers dealing with these and other questions that can be raised and answered based on the data. Our focus in this manuscript is to describe the dataset itself, providing enough information that would enable others to use the dataset in various scientific questions and from multiple perspectives.

Consistency in Units: There appears to be an inconsistency in the use of units for gamma radiation measurements. The manuscript should be reviewed to ensure that all units are used consistently and are clearly defined.
We tried to clarify this point by making clear at the beginning of section 3.2.1 that aggregation is performed by adding the gamma radiation counts measured every second into 1-minute values. Gamma radiation is presented as the number of counts per minute (the sensor we used does not provide gamma dose values).

Grammerly: Some sentences are structurally complex, which may hinder understanding. The authors are encouraged to simplify these sentences to make the article more accessible.
We tried to review and improve some sentences in the revised version of the manuscript.

Summary of main changes in the revised version:

The introduction was updated to include the contribution of electrified shower clouds to the atmospheric electric field.

Added photo of the ship in full sail, and location on the mast of the main instruments (Figure 1 in the revised version of the manuscript).

Figures 1 and 2 in the original version (Figures 2 and 3 in the revised version) were updated – changed the colours for improved clarity and correction of the colours used in Figure 3 to be consistent with Figure 2.

Figure 5 in the original version (Figure 6 in the revised version) was updated (labels added and y-axis spacing corrected).

Added photo of the simultaneous mast and shore measurements (Figure 7 in the revised version of the manuscript).

Figure 8 in the original version (Figure 10 in the revised version) was updated (changed the width of the histogram bars).

Figure 9 in the original version (Figure 11 in the revised version) was updated.

Figure 12 in the original version (Figure 14 in the revised version) was updated (negative values in solar radiation set as missing instead of zero).

The issue of assessing the reduction factor for the atmospheric electric field measurements and its impact on the accuracy of absolute values was further discussed in the revised version of the manuscript (in the introduction to section 3.1.3 and in the conclusions section).

Added to Table 2 the link to the computational notebook containing the code used for the figures in the manuscript.

Changed in Table 2 the DOI of the solar radiation computational notebook, which was updated to reflect the change of setting negative values in solar radiation as missing instead of zero. The resulting solar radiation dataset was updated, and the corresponding DOI was updated in Table 3.

Changed in Table 3 (3$^{rd}$ row) the DOI for the gamma radiation dataset. From version 2 to version 3 of that dataset a new column was added in order to have, in addition to gamma counts values, also the corresponding standard deviation. There was a mistake in that process, resulting in the loss of the latitude column. This issue was corrected in the current version of the dataset (v4). The corresponding computational notebook was updated accordingly, and therefore a new DOI is provided in Table 2 (2$^{nd}$ row).

References added: Brazenor & Harrison, 2005; Granger & Perez, 2021; Harrison, 2012; Hewlett, 1914; Kamsali et al, 2009; Liu et al, 2010; Mach et al, 2010; Mach et al, 2011; Williams & Mareev, 2014; Wilson & Cummins, 2021;